# *Arthrospira platensis* and Its Potential for Skin Regeneration in Animal Models as Support for Initiating Clinical Trials in Humans: A Systematic Review

**DOI:** 10.3390/biomedicines13092239

**Published:** 2025-09-11

**Authors:** Sara Isabel Fernández, María Estefanía Hernández, Lina Andrea Gómez

**Affiliations:** 1Faculty of Medicine, Universidad de La Sabana, Chía 140013, Cundinamarca, Colombia; sarafene@unisabana.edu.co (S.I.F.); mariahema@unisabana.edu.co (M.E.H.); 2Faculty of Medicine, Biomedical Research Center (CIBUS), Universidad de La Sabana, Chía 140013, Cundinamarca, Colombia

**Keywords:** *Spirulina*, healing, wound, rats

## Abstract

**Background/Objectives:** The search for natural alternatives to enhance wound healing has driven the investigation of bioactive compounds, such as *Spirulina*. This microalga, rich in antioxidant, anti-inflammatory, and antimicrobial properties, contains compounds like phycocyanin (C-PC), which promote cell repair, reduce inflammatory markers, and combat bacteria. Although its effects are promising, its efficacy still requires validation through human clinical trials. This article aims to review scientific publications on the use of *Spirulina* in skin regeneration using animal wound models. **Methods:** A database search was conducted for studies published between 2017 and 2024 on the effects of *Spirulina* on tissue regeneration in rats, chosen for their genetic similarity to humans. In vitro studies, those using other animal models, or studies published in languages other than Spanish or English were excluded. The review followed the PRISMA 2020 guidelines. **Results:** Four studies were analyzed, all of which demonstrated promising results in wound healing in rats. *Spirulina* was administered through oral supplements, hydrogels, and nanophytosomal formulations. These treatments accelerated wound closure and improved granulation tissue formation, vascularization, and epithelialization. Additionally, they exhibited antihyperglycemic effects in diabetic rats. **Conclusions:** The reviewed studies highlight the potential of *Spirulina platensis* to enhance wound healing, particularly in cases of diabetes and burns. Its antioxidant and anti-inflammatory properties play a crucial role in accelerating cellular regeneration and reducing inflammation, contributing to faster and more effective recovery. However, further research in humans is necessary to confirm its safety and clinical efficacy.

## 1. Introduction

The skin is the largest organ of the human body and a powerful sensory receptor that performs various essential functions, including protection against external aggressions, regulation of body temperature, absorption of ultraviolet radiation, and vitamin D metabolism. It also plays a crucial role in immune recognition and serves as an effective barrier against pathogenic microorganisms [1]. However, constant exposure to adverse environmental factors and physiological alterations makes the skin vulnerable to damage, potentially compromising its integrity and impairing its proper function [2].

Wound healing is one of the primary physiological processes of the skin. This complex biological mechanism typically occurs automatically and often goes unnoticed unless complications arise [3]. Successful wound healing requires the coordinated participation of various tissues and cell types, which contribute to the phases of hemostasis, inflammation, proliferation, and remodeling [4]. However, several factors can influence this process, including wound-specific variables, such as size, depth, wound bed condition, ischemia, infection, anatomical location, oxygen supply, pressure, and tension. Additionally, patient-related factors, such as age, nutritional status, and underlying conditions, including diabetes mellitus, peripheral vascular disease, and hyperthyroidism, can hinder proper healing, prolong wound closure time, and increase the risk of complications, such as limb amputation, sepsis, and elevated healthcare costs [5].

For these reasons, various strategies have been explored in recent years to promote wound healing. Among them are molecular therapies, such as the use of recombinant FGF7 and antibodies against MMP10, which promote epithelial regeneration and reduce tissue degradation. The peptide FOL005, derived from osteopontin, is also mentioned for its ability to enhance re-epithelialization under both normal and pathological conditions. Additionally, cellular therapies involving stem cells and bioengineered skin substitutes are explored, along with physical technologies like photobiomodulation, shock waves, and electromagnetic fields. These are complemented by innovations like smart sutures that generate electricity to stimulate healing, dressings with sensors, and portable devices for remote monitoring. All of these approaches are integrated into a holistic framework that considers not only the wound itself but also the patient’s overall condition and environment, thereby achieving faster and more effective healing [6]. One promising approach involves the use of microalgae, due to their anti-inflammatory, antioxidant, and antimicrobial properties. These characteristics help stimulate cell regeneration and accelerate the healing process. Furthermore, microalgae are rich in lipids, proteins, carbohydrates, and chlorophyll, which contribute to skin hydration and moisture retention [7,8]. Some microalgae have been identified as promising biomedical materials for tissue regeneration, particularly in dermatology [9].

Among these, *Chlorella*, a unicellular green alga found in freshwater, has been extensively studied due to its high content of protein, vitamins B12, C, and E, iron, zinc, calcium, magnesium, omega-3 fatty acids, and antioxidants. These components help combat free radical damage, support immune function, and promote cardiovascular health. On the other hand, *Spirulina*, one of the most extensively studied microalgae, has attracted special interest due to its bioactive properties that accelerate wound healing and promote cell regeneration. Encapsulated peptides derived from *Spirulina* in nanoliposomes have been shown to enhance wound contraction, epithelialization, fibroblast proliferation, collagen production, and angiogenesis [10,11].

*Spirulina* belongs to the *Arthrospira* genus. These organisms form trichomes—filamentous structures composed of cylindrical cells arranged in spirals or straight lines. The filaments typically measure 100–200 μm in length and 6–12 μm in diameter, although their size and shape may vary. The cellular structure of *Spirulina* resembles that of a typical Gram-negative prokaryotic bacterium, lacking membrane-bound organelles. Its cell wall is relatively fragile and consists of several layers, primarily composed of peptidoglycan and lipopolysaccharides. Inside of the cells are thylakoid membranes with phycobilisomes, carboxysomes, ribosomes, DNA strands, and gas vesicles. Additionally, the cells contain granules of polyglucan, polyphosphate, and cyanophycin [12].

*Spirulina platensis* has been used both pharmacologically and as a dietary supplement due to its high vitamin and protein content. One of its key biologically active compounds is phycocyanin, which distinguishes it from other microalgae. Phycocyanins are pigmented biliproteins—mainly C-phycocyanin (C-PC) and allophycocyanin—typically found in a 10:1 ratio [13]. These proteins contain linear tetrapyrrole groups (phycocyanobilin) covalently attached to specific cysteine residues. In their active form, they assemble into light-harvesting antenna complexes that help cyanobacteria capture light energy [14]. Fatty acids constitute approximately half of *Spirulina*’s total lipid content, and nearly all of its digestible carbohydrates are glucose-based polymers. Several *Spirulina* strains have had their genomes fully sequenced. The genome of *A. platensis* NIES-39 consists of a single circular chromosome approximately 6.8 million base pairs in length containing 6630 protein-coding genes, two sets of rRNA genes, and 40 tRNA genes. Another strain, *Arthrospira* PCC 8005—selected by the European Space Agency for its MELiSSA project supporting long-term space missions—has a genome of approximately 6.28 million base pairs, with a G+C content of 44.7%. It includes 5856 protein-coding genes and 176 RNA-encoding genes [15].

Phycocyanins are potent antioxidants with anti-inflammatory properties not commonly found in other microalgae, although they are characteristic of photosynthetic organisms capable of absorbing light [12,13]. Phycocyanins are classified into three main types, allophycocyanin, R-phycocyanin, and C-phycocyanin, the latter being primarily derived from *Spirulina platensis*. C-PC has been attributed regenerative effects in tissue healing due to its high hydrophilic stability within a pH range of five to eight [15,16].

Research by Madhyastha et al. (2008) [17] demonstrated that C-PC stimulates fibroblast proliferation by inducing the G1 phase of the cell cycle and increasing the expression of cyclin-dependent kinases (cdK1 and cdK2) without being cytotoxic at concentrations up to 100 μg/mL. Additionally, it promotes cell migration through the PI-3K pathway and the activation of GTPases, such as Cdc42 and Rac1, although this migration depends on the presence of urokinase-type plasminogen activator. In vivo studies in mice showed that treatment with C-PC achieved 80% wound closure within one week compared to 50% in the control group [17]. Gur et al. (2013) [18] confirmed these effects both in vitro and in vivo, observing a dose-dependent improvement in cell proliferation and migration, with the most effective dose being 33.5 μg/mL in cell cultures and 1.25% in animal models. Taken together, these findings support the use of C-PC as a promising active component in treatments for both external and internal wounds, such as ulcers [18].

Other regenerative properties include its ability to modulate inflammatory markers and influence cell remodeling. For instance, C-PC inhibits cyclooxygenase-2 (COX-2), an enzyme essential for prostaglandin synthesis via the oxidation of arachidonic acid. Because prostaglandins contribute to pain and inflammation, inhibiting COX-2 reduces the production of pro-inflammatory molecules, such as interleukins IL-1α, IL-1β, and IL-4, while increasing IL-10, an anti-inflammatory interleukin. Moreover, phycocyanins are rich in antioxidants that neutralize free radicals, thereby minimizing chronic inflammation and cellular damage. Other studies have shown that C-PC inhibits bacteria, such as *E. coli*, *B. subtilis*, and *S. pyogenes*, at concentrations of 16 μg/mL, with greater efficacy against Gram-positive bacteria. Specifically, C-PC derived from *Anabaena oryzae* exhibited IC_50_ values ranging from 30.56 to 31.25 μg/mL, comparable to or even exceeding those of benzylpenicillin. Additionally, morphological changes were observed in treated bacteria, including irregularly wrinkled and fragmented cell walls. In the case of acne, C-PC demonstrated effectiveness against *Propionibacterium acnes* and *Staphylococcus epidermidis*, with better results in water-based formulations (MIC of 1.5–1.8 mg/mL and inhibition zones up to 26.1 mm), suggesting improved compound release. These findings support the use of C-PC as a promising natural agent in the development of topical treatments for inflammatory and infectious skin conditions [14].

Cyanobacterial biomass has been considered an important source of protein in addition to its nutritional value, and it also represents a promising source of C-phycocyanin (C-PC). Cultivating cyanobacteria under controlled conditions, along with low-cost extraction processes, offers economic advantages to meet the growing demand for this pigment. Among the various factors influencing the productivity and composition of C-PC, light intensity and quality are particularly significant. To extract proteins from algal cells, specific procedures must be applied depending on the nature of the cells. However, the commercial exploitation of C-PC faces major challenges, such as its widespread use and increasing production yield. These obstacles could be overcome through closer collaboration between laboratory researchers and industrial technologists. Various strategies are being explored to reduce production and harvesting costs, as well as to evaluate new environmental conditions that favor algal cultivation. Although recent progress has been made in the production and applications of C-PC, further efforts are needed to achieve cost-effective overproduction through recombinant DNA technology. Additionally, protein engineering could enhance its nutritional and pharmacological value, thereby expanding its potential in the food, medical, and biotechnological industries [16].

This review is relevant as it addresses the growing need to identify natural, safe, and effective alternatives to promote wound healing, particularly in chronic conditions, such as diabetes and burns, where conventional treatments are often inadequate. The aim of this work is to critically evaluate recent scientific evidence on the use of *Spirulina* in wound healing, with a specific focus on rat models. These models were chosen due to their genetic, physiological, and metabolic similarities to humans, making them a widely accepted preclinical tool for translational research. By focusing exclusively on in vivo studies in rats, this review offers a more precise and meaningful assessment of *Spirulina*’s therapeutic potential, while also identifying gaps in the current literature that pave the way for future human clinical trials.

## 2. Materials and Methods

### 2.1. Literature Research

We performed an electronic search in English and Spanish in the databases PubMed, Scopus, and ScienceDirect by grouping population, context, and concept terms that constitute our question using keywords, such as *Spirulina*, healing, wounds, and animal model, to achieve the objective established for the review. The final structure of each search strategy was adjusted based on the vocabulary of the thesaurus and the Boolean operators specific to each database.

### 2.2. Inclusion and Exclusion Criteria

Articles published from the year 2017 to the year 2024 were included, in which the effect of Spirulina on tissue regeneration in wounds in animal models was studied, especially rats with incisional injuries or induced burns, with prior approval of the associated animal care and use committees. Rats are chosen as a model in numerous studies because they share approximately 90% genetic similarity with humans, suggesting that their physiology and biological responses are comparable. In fact, the mouse was the second mammal to have its genome fully sequenced, following humans in the early 2000s. The most recent genome assemblies (GRC38) show that the human genome is 3.1 Gb in size, while the mouse genome is 2.7 Gb, making it 12% smaller. Despite this difference, about 90% of both genomes can be divided into conserved syntenic regions, and 40% of human nucleotides can be aligned with those of the mouse. Additionally, there is a high degree of gene orthology: 80% of human genes and 72% of mouse genes have a one-to-one orthologous relationship, totaling 15,893 shared genes. However, the remaining genes exhibit more complex relationships or are species-specific, such as the human gene *saitohin (STH)*, which has no equivalent in mice [19,20,21].

The exclusion criteria were in vitro studies with cells of animal origin, studies with rabbits, mice, and zebrafish, studies whose objective was not to analyze the efficacy of algae like Spirulina in tissue regeneration, and articles written in languages other than Spanish or English. These exclusion criteria were considered to allow for comparison with the studies selected.

### 2.3. Screening of Studies

To reduce the risk of selection bias, relevant studies were selected in four stages. The first stage focused on eliminating duplicate records. The second stage focused on excluding articles based on title and abstract. The third stage consisted of eliminating articles that had not been performed with rat animal models, and the fourth stage consisted of full-text analysis. Inclusion and exclusion criteria were used in the last three stages. This process was based on the statements established in the PRISMA 2020 protocol.

### 2.4. Search Results

After eliminating duplicate studies and for other reasons, such as publication dates, the search retrieved 77 studies (Figure 1). Based on the information found in the titles and abstracts, 8 studies were evaluated, 4 of which were excluded. Registration information can be found in PROSPERO CRD420251004698.

## 3. Results

In this review, four scientific studies were included that investigated the role of *Spirulina platensis* in tissue regeneration. Some studies involved diabetic rats with induced lesions, while others involved healthy rats with incisional and burn lesions. The administration methods were also examined. All studies conducted macroscopic and microscopic evaluations of tissue regeneration, including histopathological analysis, studying the generation of granulation tissue, epithelialization, vascularization, inflammation, and oxidative stress. Table 1 presents the general characteristics of the selected articles.

The four studies used *Spirulina platensis* as a therapeutic agent, but in different presentations. Mehdinezha N. et al. administered spirulina and Chlorella orally to 65 diabetic Wistar rats, induced with an injection of streptozotocin at a dose of 60 mg/kg administered along with 1 cc of citrate buffer (pH = 4.5), which was prepared by slowly adding 47 mL of citric acid monohydrate solution to 50 mL of trisodium citrate dihydrate solution, a compound that injures the β cells of the pancreas. Subsequently, an 8 mm puncture wound was made on the back. The study was conducted with five groups of rats. Group I consisted of healthy rats that received a normal diet; group II included diabetic rats that also received a normal diet; in group III, the diabetic rats were fed a Spirulina diet; and group IV was composed of diabetic rats that received a Chlorella diet. Finally, group V consisted of diabetic rats that were fed a combined Spirulina and Chlorella diet [22]. Similarly, Liu T. et al. induced diabetes in Sprague-Dawley rats with an injection of streptozotocin and caused an 8 mm lesion on the back, forming six groups according to the administered treatment. In that study, they created a spirulina hydrogel where spirulina was cultivated under controlled conditions to ensure the quality of its nutrients, the biomass of spirulina was extracted, and it was mixed with polymers and gelled through temperature, pH, and gelling agents combined with other strategies, such as the use of electrical stimulation and light, which stimulate cell migration, a phenomenon known as electrotaxis, activate cellular pathways that reorganize the cytoskeleton, and increase collagen production, favoring efficient tissue repair and improving oxygenation. In a subsequent study, group allocation was determined according to the treatment administered. Group I consisted of rats treated with phosphate-buffered saline (PBS), group II included rats treated with direct electrical stimulation of the wound at a frequency of 1 Hz, with a current of 8 mA, for 30 min per session, group III comprised rats treated with Spirulina hydrogel, group IV consisted of rats treated with Spirulina hydrogel and exposed to white light at 3500 lux for 60 min per session, and group V included rats treated with Spirulina hydrogel combined with light and electrical stimulation. Finally, group VI was a comparison group treated with a hydrogel like the one made with Spirulina but without the specific concentrations to compare the efficacy of the hydrogel made for the study [23].

In the study by Refai H. et al., they used a nanophytosomal formula of Spirulina in gel, which was created using lipids (in this case, phosphatidylcholine) to form nanoparticles, which in turn form lipid vesicles that encapsulate the Spirulina extract, in addition to being incorporated into hydroxypropyl methylcellulose gels that facilitated the application and controlled release of active ingredients. In that study, they used 40 Sprague-Dawley rats with a 10 mm diameter incision on the back. The rats were organized into five groups. Group I consisted of rats without lesions, group II included rats with untreated lesions, group III comprised rats with lesions treated with MEBO^®^, a commercial local cream for lesion treatment, and group IV consisted of rats treated with *Spirulina platensis* gel. Finally, group V included rats topically treated with the nanophytosomal formula of Spirulina (SPNP-gel) [24]. Finally, at Gabeen University, polysaccharides extracted from *Spirulina platensis* enriched in a culture with constant exposure to CO_2_ were used, obtaining a water-soluble polysaccharide from Spirulina (SWSP) that was used in the healing of laser burn lesions in rats. The researchers used 18 Wistar rats that were subjected to burn lesions with a Korean-made fractional CO_2_ laser system, with controlled energy, density, and depth to generate a precise and reproducible lesion. The lesion area was 2 cm^2^ on the back. The rats were divided into three groups. Group I consisted of rats treated with a 30% glycerol solution, group II included rats treated with Cytolcentella, the standard treatment for lesions, which contains Centella asiatica extract as the active component, and group III comprised rats treated with SWSP [25].


Healing Speed and Closure Percentage


In the four studies, variations in healing speed or closure percentage were evidenced according to the type of treatment. In the study by Mehdinezha N. et al., using Digimazer image software, it was demonstrated that in the group treated with Spirulina and Chlorella, there was greater formation of granulation tissue on days 3, 7, 14, and 21 (*p* = 0.02), on day 14 (*p* = 0.03), and on day 21. Additionally, in all groups where microalgae were used, the lesions were healed at 100% by day 21, except in the diabetic control group, where 97.8% healing was achieved [22].

In the study by Liu T. et al., evaluated using image software (ImageJ version 9.50); National Institutes of Health (NIH), it was found that the wound area of the group treated with Spirulina hydrogel was reduced by 58.3% compared to its original size on day 7 (*p* = 0.01). The group treated with Spirulina hydrogel and electrical stimulation showed increased closure, with the wound area reduced to 18.8% on day 10 (*p* = 0.001) compared to the control group, which still had healing difficulties after day 10 [23].

On the other hand, in the study conducted by Refai H. et al., healthy rats had complete closure in 14 days with increased collagen and reduced inflammation (*p* < 0.001) [24]. At Gabeen University, using image software (Auto CAD RL 14), it was evidenced that lesions treated with SWSP showed a red coloration that evolved to dark brown in one day and persisted until the third day. On the fifth day, a scab was detected that disappeared on the seventh day, allowing for the formation of a new pink epithelium that completely covered the lesion compared to the other groups, where there was no evidence of complete healing of the lesions. The lesion’s closure was completely achieved in the group treated with SWSP, reaching an area of 0.05 cm compared to 0.25 cm in the group treated with Cytolcentella at the end of the study. Epithelialization was complete in 7 days, with increased hydroxyproline synthesis (*p* < 0.001) [25]. Table 2 presents a comparison of the wound area reduction percentages reported in each article.


Cellular and Biochemical Effects


In the study by Mehdinezha N. et al., a biochemical evaluation was performed with blood taken on day 21. Glucose was measured with a glucometer, and vascular endothelial growth factor (VEGF) and high-sensitivity C-reactive protein (hs-CRP) were measured using the ELISA method. In the histopathological study, granulation, vascularization, and epithelialization were evaluated. In the group treated with Spirulina and Chlorella, there was greater vascularization and epithelialization compared to the control group, which was slower (*p* = 0.05). Additionally, they observed a decrease in blood glucose levels and weight in those rats treated with microalgae. The researchers concluded that the use of oral supplementation of Spirulina and Chlorella, both individually and in combination, had a positive impact on wound healing by improving the processes of granulation tissue formation, vascularization, and epithelialization, recommending this treatment in the management of diabetic ulcers [22].

On the other hand, in the study by Liu T. et al., minimal infiltration of inflammatory cells, increased collagen deposition, a 452% increase in microvessel density, a 112% increase in VEGF expression (*p* < 0.001), increased cell proliferation, and inhibition of apoptosis by decreased caspase-3 expression (*p* < 0.001) were found. Reactive oxygen species (ROS) levels were reduced by 75%, and tumor necrosis factor-alpha (TNF-a) and interleukin 1B (IL-1B) levels were reduced. Increased levels of interleukin 10 (IL-10) and transforming growth factor-beta (TGF-B) were also observed [23].

In the study by Refai H. et al., an anti-inflammatory effect was shown with a decrease in high mobility group box 1 (HMGB-1) proteins, which are associated with an elevated inflammatory response and whose expression decreased with treatment, confirming the anti-inflammatory effect of SNPN. They also demonstrated the antioxidant effect with a marked increase in nuclear factor erythroid 2 (Nrf-2) and heme oxygenase-1 (HO-1), essential proteins for cellular protection against oxidative stress. Additionally, they found decreased levels of caspase-3 and apoptosis-inducing factor (AIF), showing an anti-apoptotic effect, which is inferred to promote cell survival and faster wound healing [24].

At Gabeen University, in histological plates, complete epithelialization with well-structured cell layers without cellular inflammation was found in the group treated with SWSP. Additionally, a significant decrease in inflammatory cells with anti-inflammatory activity was evidenced. Therefore, the researchers concluded that the healing effect of SWSP could be due to mechanisms related to increased epithelialization and neovascularization rates, scavenging of destructive free radicals, reduction of inflammation, and infection control, which could be due to the antioxidant, anti-inflammatory, and antimicrobial components of *Spirulina platensis* polysaccharide [25]. Table 3 presents a comparison of the results based on *p*-values for each parameter evaluated across the different articles, with the aim of determining the effectiveness of wound treatment.

## 4. Discussion

The skin is a vital organ for human survival and well-being due to its physical and immunological barrier functions, which protect against microorganisms and UV radiation, as well as its roles in temperature regulation and vitamin D synthesis. To perform these functions effectively, the skin’s structural integrity must be maintained. However, various diseases can compromise skin function by impairing proper perfusion. Consequently, numerous strategies have been explored in recent years to accelerate the healing of skin lesions, among which the use of microalgae, such as *Spirulina platensis,* has gained attention.

This review compared and analyzed studies on the use of *Spirulina platensis* in treating skin lesions in rat models. Four articles were reviewed, evaluating macroscopic characteristics of the lesions, wound closure percentage, histological features, such as granulation tissue formation, vascularization, and epithelialization, as well as biochemical and immunohistochemical parameters.

The findings demonstrate that *Spirulina platensis* and its various formulations offer significant benefits in skin wound healing. The reviewed studies reported improvements in granulation tissue formation, vascularization, epithelialization, and high wound closure rates, particularly in diabetic models. Additionally, oral supplementation with *Spirulina* and *Chlorella* showed a remarkable ability to reduce blood glucose levels through several mechanisms. These include reducing oxidative stress via antioxidants that protect pancreatic beta cells and enhance their insulin production capacity, increasing peripheral tissue sensitivity to insulin through bioactive compounds like phycocyanin and regulating key enzymes involved in glucose metabolism. Furthermore, *Spirulina* exerts anti-inflammatory effects by reducing systemic inflammation associated with insulin resistance. These combined actions resulted in significant reductions in blood glucose levels and body weight, suggesting a complementary antihyperglycemic potential for managing diabetic patients [22].

In comparison, innovative techniques, such as hydrogels infused with *Spirulina* and combined with electrical stimulation, offer advantages over conventional treatments. This approach not only accelerates healing but also promotes cell migration and angiogenesis through the activation of voltage-gated ion channels [23]. Similarly, nanophytosomal gels have demonstrated efficient delivery of active compounds, representing a significant advancement over traditional topical formulations [24].

One of the main strengths of these studies is the diversity of methodological approaches used to evaluate the effectiveness of *Spirulina*, including macroscopic, histological, biochemical, and immunohistochemical analyses. However, these studies were limited to animal models, which restricts the direct extrapolation of results to human clinical scenarios.

In addition, in vitro studies have also been conducted. One such study, carried out in 2015 by Syarina PNA and collaborators, evaluated the effects of *Spirulina* extracts on the migration and proliferation of human dermal fibroblasts using an in vitro scratch assay, along with the identification of bioactive compounds responsible for the observed activity. Allantoin was used as a positive control, and images were taken after 24 h. The aqueous extract promoted cell migration and proliferation comparable to allantoin, whereas the methanolic and ethanolic extracts showed no significant effects. The aqueous extract was found to contain compounds, such as hydrolyzed dynamic acid, naringenin, kaempferol, isomeric derivatives of phosphatidylserine, and sulfoquinovosyl diacylglycerol, known for their anti-inflammatory, antioxidant, immunomodulatory, and pro-coagulant properties [26].

Another example is a study conducted by Sevimli-Gür C, which aimed to identify and characterize the wound-healing effects of C-phycocyanin, a compound isolated from *Spirulina platensis*. The effects on cell viability and proliferation were assessed using the MTT assay in primary human skin epithelial cells. Wound healing was evaluated using an in vitro model in which a 5 mm circular area was cleared of cells to simulate an artificial wound. Both the crude extract of *Spirulina* and C-phycocyanin demonstrated enhanced wound-healing activity at a concentration of 33.5 µg/mL. Additionally, the in vivo efficacy of both treatments was evaluated in male Sprague-Dawley rats. Histological analyses assessed re-epithelialization, neovascularization, inflammatory cell presence, granulation tissue formation, and its maturation. The study concluded that C-phycocyanin at a concentration of 1.25% exhibited superior wound-healing effects by day 7 compared to other formulations [27].

Moreover, other biological effects, such as antitumor activity, have also been explored. A 2012 study evaluated the effects of C-phycocyanin (C-PC), an antioxidant pigment derived from *Spirulina platensis*, on TPA-induced tumor promotion in mouse skin. The study focused on key molecular markers related to cell proliferation, inflammation, and apoptosis. Swiss mice received a topical application of TPA (10 µg) to induce tumor promotion, followed by treatment with C-PC at doses of 50, 200, or 400 µg. The results showed that C-PC inhibited the expression of ornithine decarboxylase (ODC) at both the protein and gene levels and restored and enhanced the expression of transglutaminase 2 (TG2), suggesting a pro-apoptotic effect. It also significantly reduced the overexpression of COX-2 and IL-6, demonstrating a clear anti-inflammatory effect. Furthermore, it inhibited phosphorylated STAT3, thereby disrupting pro-tumor signaling pathways [28].

These findings underscore the need for clinical trials to confirm the safety and efficacy of *Spirulina*-based treatments in humans.

The evidence suggests that *Spirulina* could be a promising solution for the treatment of burns and chronic wounds, particularly in diabetic patients, due to its antioxidant, anti-inflammatory, and antimicrobial properties, as well as its potential as a natural chemo preventive agent, especially in the context of inflammation-induced skin cancer. These qualities may reduce the risk of infection and improve both the quality and speed of wound healing and chemopreventive agents [25].

## 5. Conclusions

Overall, these studies underscore the potential of *Spirulina* platensis and other microalgae-derived compounds as effective treatments that enhance tissue regeneration and injury healing in animal models, especially in the context of diabetic and burn injuries. Studies support its potential as a safe and effective treatment to accelerate healing of acute injuries, as it possesses antioxidant and anti-inflammatory properties and stimulates collagen production, projecting these results for future research and clinical applications. The variability in oral, topical administration methods, combined with electrical stimulation, opens multiple avenues to optimize its use in different types of injuries and clinical conditions. However, further research is needed to confirm the effectiveness of Spirulina in human clinical trials. We hope that this review will pave the way for the development of new Spirulina-based treatments, with the aim that these advances will translate into therapies that improve patients’ quality of life and accelerate skin regeneration.

## Figures and Tables

**Figure 1 biomedicines-13-02239-f001:**
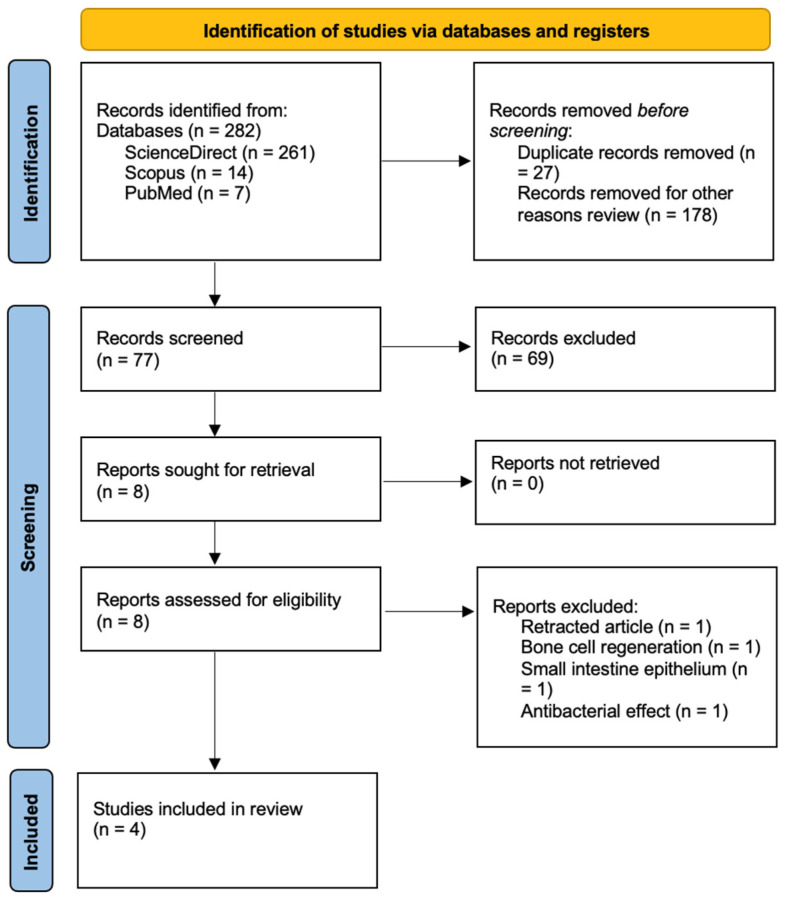
PRISMA flow chart.

**Table 1 biomedicines-13-02239-t001:** General characteristics of the selected articles.

Principal Author	Year	Objective	Population	Lesion	Results
Mehdinezha, et al. [22]	2021	Evaluate the effect of Spirulina and Chlorella alone and in combination on wound healing in diabetic rats, analyzing their impact on tissue regeneration, inflammation, and glucose control.	Included 65 experimental Wistar rats. Induction of diabetes through injection of streptozotocin 60 mg/kg plus 1 cc citrate buffer; rats were anesthetized for wound induction and randomly divided into five groups.	The wound was made with an 8 mm diameter punch on the dorsum. The dermis and epidermis were excised.	Spirulina and Chlorella significantly improved wound healing in terms of granulation, vascularization, and epithelialization. Blood sugar levels were reduced. There was no significant effect on hs-CRP and VEGF levels.
Liu, et al. [23]	2024	Evaluate a therapeutic strategy combining hydrogels capable of generating oxygen using *Spirulina platensis* and electrical stimulation to improve the healing of chronic wounds in diabetic patients.	An unknown number of experimental Sprague-Dawley rats. Rats were given diabetes through injection with streptozotocin and then anesthetized, given wounds, and then randomly assigned into six groups to receive treatment.	The wound was made with an 8 mm diameter punch on the dorsum. The dermis and epidermis were excised.	It was shown to alleviate hypoxia, promote healing, and reduce inflammation. Hydrogel released oxygen in a sustained manner due to the photosynthetic activity of Spirulina. Significantly accelerated healing, improving tissue formation and neovascularization and reducing inflammation and cell apoptosis. Decreased oxidative stress.
Refai, et al. [24]	2023	Evaluate the efficacy of a nanofitosomal formulation loaded with dried hydroalcoholic extract of *Spirulina platensis* in promoting incisional wound healing in rats.	Total of 40 experimental Sprague-Dawley rats. Rats were anesthetized and subsequently given 10 mm wounds and randomly divided into 5 groups.	A 10 mm diameter incisional wound on the dorsum and full thickness (dermis and epidermis).	Faster and more complete closure of the lesions, achieving total epithelialization by day 14, surpassing the other treatments. Reduced inflammatory markers through upregulation of HMGB-1. Stimulated autophagy and reduced apoptosis more effectively than other treatments. Promoted a marked increase in VEGF expression and collagen synthesis, improving the quality of scar tissue.
Aydi, et al. [25]	2022	Investigate the structural characteristics and physicochemical properties of polysaccharides extracted from *Spirulina platensis* grown under CO_2_-enriched conditions. In addition, their antioxidant and cytotoxic activities, as well as their efficacy in laser burn wound healing in rats, were evaluated.	Total of 18 experimental Wistar rats. The rats were anesthetized and then given a 2 cm^2^ laser wound, divided into three groups.	CO_2_ fractional laser burn wound (DSE, chorea) of 2 cm^2^, partial thickness.	It did not generate toxicity. Significantly reduced wound size. Increased hydroxyproline content associated with rapid collagen synthesis and fibroblast proliferation. Complete epithelialization, absence of cellular inflammation, and formation of skin appendages. Reduction of inflammatory cells.

**Table 2 biomedicines-13-02239-t002:** Percentages of wound area reduction.

Principal Author	Final Area %	Reduction %
Mehdinezha, et al. [22]	0.0%	100%
Liu, et al. [23]	3.9%	96.1%
Refai, et al. [24]	0.0%	100%
Aydi, et al. [25]	0.05%	95.0%

**Table 3 biomedicines-13-02239-t003:** *p*-values obtained according to the variables of granulation tissue formation, vascularization, epithelialization, and percentage of wound healing.

Principal Author	Granulation Tissue Formation *p* Value	Vascularization *p* Value	Epithelialization *p* Value	Percentage of Wound Healing *p* Value
Mehdinezha, et al. [22]	0.003	0.053	0.053	0.001
Liu, et al. [23]	-	<0.001	<0.001	<0.005
Refai, et al. [24]	<0.05	<0.05	<0.05	<0.001
Aydi, et al. [25]	<0.05	<0.05	<0.05	<0.05

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
