# Peer review of "Arthrospira platensis and Its Potential for Skin Regeneration in Animal Models as Support for Initiating Clinical Trials in Humans: A Systematic Review"

_biomedicines, 2025, doi:10.3390/biomedicines13092239_

Round 1
Reviewer 1 Report
Comments and Suggestions for Authors
- Abstract: The skin, the largest organ, performs vital functions such as protection, thermal 12 regulation, vitamin D production, and serving as an immune barrier. Write serves in place of serving.
- Why was Spirulina, a microalga, selected instead of several other available items? Justification is required in the manuscript.
- Time span for literature review is very short-(2017-2024)
- In conclusion, the studies emphasize the potential of Spirulina platensis 31 to enhance wound healing, especially in cases of diabetes and burns, due to its antioxidant 32 and anti-inflammatory properties.---rewrite the sentence
- Line 41 and 45- some time full stop is after reference and some time before reference. Justify the reason.
- Reference is to be in large bracket .
- Line 69-One of the most studied algae is Spirulina, a seaweed belonging to the Arthrospira family. How it can be concluded that it is most studies as there is no reference and no other line of text for the same.
- Line 89-in vitro – it must be italics
- Line 109 in vitro
- After eliminating duplicate studies and for other reasons such as publication dates, the search retrieved 255 studies
- Way of writing result starting from researcher's name is not ok
- Line 136: groups of animals must be written in line form.
- Molecular approaches behind its mechanism and future scope is missing in the manuscript.
Significant language correction is required
Author Response
- Abstract: The skin, the largest organ, performs vital functions such as protection, thermal 12 regulation, vitamin D production, and serving as an immune barrier. Write serves in place of serving: We replace the word, it is underlined in red on page number 1, line 14
- Why was Spirulina, a microalga, selected instead of several other available items? Justification is required in the manuscript: We chose spirulina because it is a microalga that is currently being studied and has been shown to have a significant impact on tissue regeneration, stimulating cell proliferation and oxygenation, which are fundamental factors for healing. Additionally, being a biological material, it is low-cost and easily adapts to different forms of application. it is underlined in red on page number 2, line 71
- Time span for literature review is very short-(2017-2024): The chosen time frame was selected because research on spirulina in tissue regeneration is 'new' and there are few articles in which animal trials have been conducted. This is why we chose this time period.
- In conclusion, the studies emphasize the potential of Spirulina platensis 31 to enhance wound healing, especially in cases of diabetes and burns, due to its antioxidant 32 and anti-inflammatory properties.---rewrite the sentence: We corrected the idea and it is now underlined in red on page 1, lines 33-36
- Line 41 and 45- some time full stop is after reference and some time before reference. Justify the reason: We identified the error; there is no reason, it is just a typing mistake.
- Reference is to be in large bracket: We identified the error and corrected it in each of the references
- Line 69-One of the most studied algae is Spirulina, a seaweed belonging to the Arthrospira family. How it can be concluded that it is most studies as there is no reference and no other line of text for the same: We used the wrong expression; it is not the most studied alga, but it is one of the most studied in the field of tissue regeneration in recent years, along with chlorella, it is underlined in red on page number 2, line 71
- Line 89-in vitro – it must be italics: We identified the error, it is underlined in red.
- Line 109 in vitro: We identified the error, it is underlined in red.
- After eliminating duplicate studies and for other reasons such as publication dates, the search retrieved 255 studies: After removing the duplicate articles, which totaled 27, we also eliminated all studies that were not animal or cell trials, removing 178 in total. Finally, our search resulted in 77 articles. we identified the error, it is underlined in red, in page 3, line 130
- Way of writing result starting from researcher's name is not ok: We rewrote the results and organized them according to the main findings that we consider important for the objective of our review. it is underlined in red, in page 4 - 7
- Line 136: groups of animals must be written in line form: We identified the error and rewrote the animal groups in a linear format. it is underlined in red, in page 4-5
- Molecular approaches behind its mechanism and future scope is missing in the manuscript: In our review, we attempted to include all biomolecular factors related to the action of spirulina in tissue regeneration that we found in the review articles. However, we did not include additional biomolecular information on the generation of application mechanisms because we consider it to be somewhat outside the scope of our review.
We would like to express our sincere gratitude for your valuable corrections to our article. Your insights and suggestions have greatly improved the quality and clarity of our work.
Reviewer 2 Report
Comments and Suggestions for Authors
The manuscript addresses a relevant topic: the potential of Spirulina platensis in wound healing and skin regeneration. The subject is of growing interest, however, the manuscript exhibits several structural, methodological, and scientific weaknesses that require revision before publication in a peer-reviewed journal like Biomedicines.
Inclusion and exclusion criteria are questionable. Most animal models are excluded (pig for example?) as well as cell lines usually used for wound healing studies, such as rabbit corneal epithelium, for example.
Moreover, English language should be revised because there are several grammar issues and the meaning of many of the sentences is not clear.
The article reviews four animal studies and the discussion remains descriptive rather than critical or comparative. An analysis of either the quality of the included studies or divergent results is not performed.
Comments on the Quality of English LanguageEnglish language should be revised because there are several grammar issues and the meaning of many of the sentences is not clear.
Author Response
We would like to know more in-depth about the structural, methodological, and scientific deficiencies you find in our review article. We are very interested in addressing them to produce valuable material. Regarding the inclusion criteria, we aimed to focus on a specific species to compare the results more easily and to create a clear conclusion to determine whether spirulina is a biological material with a truly positive impact on tissue regeneration. On the other hand, being a new material under study, there are not many articles in which it has been tested with other animal models or human cell models, considering that our objective is to encourage its applicability in humans. Regarding the English language, we will request support provided by the journal. Additionally, we modified the discussion section where we conducted a deeper analysis of the studies and their potential applicability in the future.
We would like to express our sincere gratitude for your valuable corrections to our article. Your insights and suggestions have greatly improved the quality and clarity of our work.
Reviewer 3 Report
Comments and Suggestions for Authors
Comments
The article is a review of scientific articles on the topic of "Spirulina Platensis and its potential for skin regeneration in animal models."
There are comments and questions about the text of the article.
- The main comment to the authors is that the article contains too superficial a presentation of the material on the topic. In research articles, as well as in the review "Breschi A, Gingeras TR, Guigó R. Comparative transcriptomics in humans and mice. Nat Rev Genet. 2017 Jul; 18 (7): 425-440. doi: 410 10.1038/nrg.2017.19. Epub 2017 May 8. PMID: 28479595; PMCID: PMC6413734", which the authors discuss in the text of this review, there is quite a lot of material. The authors of this article provide only generalizations of the results, and there is absolutely no illustrative material. In my opinion, such material is necessary for a more complete understanding of the stated positions of the article. In particular, it is possible to provide figures from the original articles demonstrating the dynamics of wound healing depending on the drugs used, figures with the dynamics of other parameters, for example, glucose depending on the stage of healing, vascular density, etc. These processes are very briefly described in this article. Supplementing the article with figures and tables would allow us to get a more complete idea of ​​the potential of the claimed medicinal properties of Spirulina Platensis. This is all the more necessary because the article is very small in volume for a review. PRISMA recommendations do not reject the use of tables, figures and other illustrative materials.
- In my opinion, the phrase in the abstract "The skin, the largest organ, performs vital functions such as protection, thermal regulation, and vitamin D synthesis, and serves as an immune barrier. Its healing capacity depends on complex processes such as hemostasis, inflammation, proliferation, and remodeling, which can be affected by factors like wound size and depth, infections, oxygenation, age, and diseases like diabetes» should be removed, since the abstract usually indicates the results of the work conducted by the authors, and it almost verbatim coincides with the phrase in the introduction «The skin is the largest organ of the human body and a powerful sensory receptor that performs various essential functions, including protection against external aggressions, regulation of body temperature, absorption of ultraviolet radiation, and vitamin D metabolism. It also plays a crucial role in immune recognition and serves as an effective barrier against pathogenic microorganisms [1]. However, constant exposure to adverse environmental factors and physiological alterations makes the skin vulnerable to damage, potentially compromising its integrity and impairing its proper function [2].
- In my opinion, the introduction could have paid more attention to various strategies for promoting wound healing. The small volume of the review allows for this.
- Authors of the article should be careful when coding words. In some cases, for example for the phrases "R-phycocyanin (R-PC), phosphate-buffered saline (PBS) and some others, the abbreviation is introduced, but is not used further in the text. Why is this abbreviation necessary? Usually, an abbreviation is introduced when the phrase is used more than once. It is necessary to carefully read the text and remove unnecessary abbreviations.
Author Response
- In the current revision of our review, we included more information about the articles selected, added specific results from those studies, and created tables to present the findings more clearly, allowing for a more visual comparison of the selected articles.
- We removed the indicated sentence from the abstract and restructured it accordingly.
- We included the various existing strategies currently available to promote wound healing.
- We identified the error and corrected it throughout the article.
All corrections were highlighted in red.
We are very grateful for the corrections made and for the time you invested in them
Reviewer 4 Report
Comments and Suggestions for Authors
This article presents the review of scientific publications (2017 – 2024) on the use of Spirulina platensis for skin regeneration in rats selected based on their genetic similarity to humans. In vitro studies, studies using other animal models, or studies in languages ​​other than Spanish or English were excluded according to PRISMA 2020 guidelines. The review analyzed four articles with promising results. Spirulina used in oral supplements, hydrogels and nanophytosomal formulations accelerates lesion closure, improved granulation tissue formation, vascularization and epithelialization, and demonstrates antihyperglycemic effects in diabetic rats.
The present article is within the scope of the journal Biomedicines. However, the authors are encouraged to revise the manuscript with respect to the following points:
Point 1: Page 1, Line 1 - The article should be classified as a Systematic review.
Point 2: Page 1, Lines 2-4 - Change the title for “Spirulina Platensis and Its Potential for Skin Regeneration in Animal Models as Support for Initiating Clinical Trials in Humans: A Systematic Review”
Point 3: Article should have a structured abstract of around 250 words and contain the following headings: Background/Objectives, Methods, Results, and Conclusions.
Point 4: Page 2-3, Lines 86-93 – Include relevant references.
Point 5: Page 3, Lines 94-96 - Include relevant reference.
Point 6: Page 3, Lines 96-100 - Include relevant reference.
Point 7: Page 3, Lines 119-122 - It is necessary to emphasise why this review is important and to re-formulate the purpose of the work with a focus on rats models selected based on their genetic similarity to humans, as well as its novelty and significance.
Point 8: Page 5, Move Table 1 to the Results section before the text with detailed descriptions of the articles (after lines 158-165). Remove the third column from Table 1. Rearrange the columns in the following order: Author, Year, Objective, Population, Lesion, Results.
Point 9: Page 5, Line 167 - Check the molarity of the citrate buffer.
Point 10: Page 5, Lines 166-174 – Invalid reference specified.
Point 11: Page 5, Line 183 – Remove the typo.
Point 12: Page 5, Line 193 – Remove the typo.
Point 13: Page 6, Line 203 – Invalid reference specified.
Point 14: There are a lot of typos in the article.
Author Response
- The article was reclassified as a systematic review
- We changed the title of the article
- The abstract was restructured as requested
- 5. 6. References were organized, indicating the specific fragments they correspond to
7. The indicated paragraph was corrected and is highlighted in red in lines 163–172
8. The table was moved to the specified location, and the columns were reorganized accordingly
9. The molarity of the citrate buffer was modified, as highlighted in red in lines 230–233
10. The reference highlighted in red on line 230 was corrected
13. The reference highlighted in red on line 268 was corrected
11. 12. 14. Typographical errors throughout the article were corrected
We are very grateful for the corrections made. We truly appreciate your time and willingness to support us
Round 2
Reviewer 1 Report
Comments and Suggestions for Authors
- Significant improvement has been done but still some points are still to be addressed
Molecular approaches behind its mechanism and future scope are missing in the manuscript this time again.
Without a future scope, how will a new researcher find out the research gap between existing research and already done research? It must be added.
In brief, the molecular mechanism must be added to exhibit the scientific interest to readers.
- Animals grouping is again mentioned in bullet type; however, it is again recommended to write in running text form
- Reference is to be in large bracket [Ref. No.]. Again the errors persist.
- Time span for literature review is very short-(2017-2024)..
The response given that The chosen time frame was selected because research on spirulina in tissue regeneration is 'new' and there are few articles in which animal trials have been conducted. This is why we chose this time period.
There are number of literature on the said came. Add these.
https://pmc.ncbi.nlm.nih.gov/articles/PMC4800779/
https://www.thieme-connect.com/products/ejournals/abstract/10.1055/s-0029-1234384
https://www.sciencedirect.com/science/article/abs/pii/S1382668912001172
Author Response
1: Molecular approaches behind its mechanism and future scope are missing in the manuscript this time again.
We have added molecular approaches in the introduction, specifically between lines 76 and 100. Please let us know if this is what you were referring to in your correction.
2. Animals grouping is again mentioned in bullet type; however, it is again recommended to write in running text form
We described the groups in running text format; this section is underlined in red in the results section.
3. Reference is to be in large bracket [Ref. No.]. Again the errors persist.
We corrected the references in large brackets.
4. There are number of literature on the said came. Add these.
Regarding our response about the limited number of articles, we were referring specifically to the lack of studies that focus solely on tissue regeneration in wounds in rat models. Our intention was to ensure a proper comparison of results within a highly similar context, allowing us to draw more accurate and meaningful conclusions. The articles provided primarily focus on in vitro studies rather than in vivo models.
We included these studies in the discussion section because they support the overall aim of the review, which is to encourage more in vivo research in humans. We would like to know whether this correction is sufficient, or if we should also include these studies in the results section and add them to our table, even though they are not exclusively focused on wound healing in rats.
We would like to express our sincere gratitude to the reviewers and editors for their valuable comments and suggestions, which have significantly improved the quality and clarity of our manuscript. Your insightful feedback helped us refine our arguments, strengthen our methodology, and enhance the overall presentation of our findings. We appreciate the time and effort dedicated to reviewing our work and are grateful for the opportunity to revise and improve our article accordingly.
Reviewer 4 Report
Comments and Suggestions for Authors
The authors have made all the changes requested by the reviewer.
However, the article needs further correction before acceptance.
Point 1: Line 224: Include a sentence that Table 1 presents "General characteristics of the selected articles".
Point 2: In table 1, in the first column with "Principal Author", to include a reference number after the last name.
Point 3: In tables 2 and 3, change the "Article" column to "Principal Author" with a reference number as table 1.
Point 4: The authors have added new tables 2 and 3 in the revised text. However, there is no links to these tables in the text of the article. It is necessary to explain in the text the data presented in tables 2 and 3. Check the complete correspondence between the table data and the text.
Author Response
Point 1: A sentence was added in line 224 stating that Table 1 presents the “General characteristics of the selected articles,” in order to properly connect the table to the text. This correction was highlighted in red.
Point 2: The corresponding references were added next to the surname of the principal author in the first column of Table 1, titled “Principal Author.” This modification was also highlighted in red.
Point 3: The column headers in Tables 2 and 3 were revised, replacing “Article” with “Principal Author” and including the appropriate reference number, consistent with Table 1. These changes were highlighted in red.
Point 4: The placement of Tables 2 and 3 within the text was reorganized, and an explanation was added to clarify their purpose as a comparative tool for the reported results. These revisions were clearly highlighted in red for easy identification.
Round 3
Reviewer 1 Report
Comments and Suggestions for Authors
Significant changes are done in the manuscript by authors. Now it may be accepted.
Comments on the Quality of English Languageok
Author Response
We are very grateful for the corrections made and for the time you invested in them.